# Caffeine Intake Alters Recovery Sleep after Sleep Deprivation

**DOI:** 10.3390/nu16203442

**Published:** 2024-10-11

**Authors:** Benoit Pauchon, Vincent Beauchamps, Danielle Gomez-Mérino, Mégane Erblang, Catherine Drogou, Pascal Van Beers, Mathias Guillard, Michaël Quiquempoix, Damien Léger, Mounir Chennaoui, Fabien Sauvet

**Affiliations:** 1Ecole du Val de Grace, 75006 Paris, France; benoit.pauchon@gmail.com; 2Hôpital d’Instruction des Armées (HIA) Sainte Anne, 83100 Toulon, France; 3Institut de Recherche Biomédicale des Armées (IRBA), 91190 Brétigny sur Orge, France; vincent.beauchamps@outlook.com (V.B.);; 4URP 7330 VIFASOM, Université Paris Cité, Hôtel Dieu, 75004 Paris, France; 5UMR LBEPS, Université Paris Saclay, 91000 Evry, France; 6APHP, Hôtel-Dieu, Centre du Sommeil et de la Vigilance, DMU THOROS, 75004 Paris, France

**Keywords:** sleep deprivation, caffeine intake, sleep continuity, sleep stability, sleep organization, sleep density, EEG

## Abstract

Background: Caffeine is a well-known psychostimulant reputed to alleviate the deleterious effects of sleep deprivation. Nevertheless, caffeine can alter sleep duration and quality, particularly during recovery sleep. We evaluated the effects of acute caffeine intake on the duration and quality of recovery sleep following total sleep deprivation (TSD), taking into account daily caffeine consumption. Methods: Forty-one participants performed a double-blind, crossover TSD protocol (38 h of continuous wakefulness) with acute caffeine or placebo. Caffeine (2.5 mg/kg) or placebo was administered twice during continuous wakefulness (last treatment 6.5 h before bedtime for the recovery night). Polysomnographic measurements were recorded using a connected headband. Results: TSD was associated with a rebound in total sleep time (TST) on the recovery night (+110.2 ± 23.2 min, *p* < 0.001). Caffeine intake decreased this recovery TST (−30.2 ± 8.2 min *p* = 0.02) and the N3 sleep stage duration (−35.6 ± 23.2 min, *p* < 0.01). Caffeine intake altered recovery sleep continuity (increased number of long awakenings), stability (higher stage transition frequency), and organization (less time spent in complete sleep cycle) and decreased the delta power spectral density during NREM sleep. On the recovery night, habitual daily caffeine consumption was negatively correlated with TST in caffeine and placebo conditions and positively correlated with wake after sleep onset (WASO) duration and with the frequency of long (>2 min) awakenings in the caffeine condition only. Conclusions: Acute caffeine intake during TSD affects nighttime recovery sleep, with an interaction with daily consumption. These results may influence advice on caffeine intake for night-shift workers. (NCT03859882).

## 1. Introduction

About 10 to 25% of workers around the world [1] are exposed to night-shift work, with potentially deleterious consequences for their health [2]. Prolonged wakefulness induced by night-shift work results in an increased risk of accidents and medical or professional errors associated with increased daytime sleepiness and decreased cognitive performance [3]. It has been demonstrated that, after 36 h of sleep deprivation, at least 9 to 10 h of sleep is required to recover cognitive functions [4].

Caffeine is usually proposed to manage sleep-loss-related neurobehavioral impairments, as it is known to decrease daytime sleepiness and increase alertness [5,6]. Caffeine is a non-selective antagonist of adenosine receptors (mainly A1 and A2A), which modulate glutamatergic, cholinergic, dopaminergic, serotoninergic, and noradrenergic neurotransmission [7]. During sleep deprivation in a laboratory study, acute caffeine administration (200 to 600 mg per day) helped to reduce sleepiness and EEG theta activity during wakefulness [6,8,9] and to protect cognitive function by preventing a decrease in sustained attention [10,11].

The sleep-disruptive effects of caffeine administration at bedtime are also well documented [12], and caffeine administration has been used as a model of insomnia [13,14]. Dose–response studies have demonstrated that caffeine administered at or near bedtime is associated with significant sleep disturbances [13,15]. Indeed, a moderate dose of caffeine (i.e., 100–400 mg) at bedtime, or 3 to 6 h prior to bedtime, decreased total sleep duration and sleep quality [16,17], increased sleep latency, and mainly reduced the lowest EEG delta band (0.25–0.5 Hz) in non-rapid eye movement (NREM) sleep [17]. Moreover, in REM sleep, the EEG power density was suppressed in the frequency range of 0.75–4.5 and 5.25–6.0 Hz [17].

In addition, acute caffeine intake in response to insufficient sleep can interfere with the onset and maintenance of the subsequent restorative sleep necessary for the recovery of physical and cognitive performance [18,19,20]. In two studies, the characteristics of the following nighttime recovery sleep were investigated, and it was found that the ingestion of 100–400 mg of caffeine did not change the sleep-deprivation-related increase in total sleep time (TST), sleep efficiency, or slow-wave sleep (SWS) or the related decrease in sleep latency, stage 1 sleep, and wakefulness after sleep onset (WASO) [18,20]. The only caffeine effect was on EEG power spectra in NREM sleep during the first 480 min of recovery sleep, with caffeine attenuating the sleep-deprivation-related increase in power in the 0.75–2.0 Hz range and the related reduction power in the entire 11.25–20.0 Hz band [18]. Furthermore, Bonnet et al. [21] observed that 400 mg of caffeine immediately prior to 3.5 h of nighttime sleep (starting at 02:00 h) increased stage 1 sleep and decreased stage 4 sleep compared to placebo, without a subsequent caffeine effect on sleep macrostructure after 41 h of sleep deprivation. Wesensten et al. [19] described no differences between caffeine (600 mg) and placebo (taken around 21 h before bedtime) on TST during the night following 85 h of wakefulness, but they did not compare this with the baseline night. The aforementioned studies point out that the conditions under which caffeine is administered (e.g., the amount ingested and the last time of intake) may explain the lack of caffeine effect on the macrostructure of nighttime sleep recovery following sleep deprivation. Indeed, the last caffeine intake before going to sleep after sleep deprivation was around 14 h in the studies by Landolt et al. [18] and Beaumont et al. [20].

With respect to acute caffeine effects on recovery sleep following prolonged wakefulness, several authors have investigated the following daytime recovery sleep, and the effects of caffeine are described as more marked on daytime recovery sleep than on nocturnal sleep [22,23].

The question of the effects of caffeine on recovery sleep is important in the context of the increasingly widespread use of caffeinated energy drinks and the caffeine content of domestic coffee [24]. Regular caffeine consumption during the day also alters sleep, particularly in the afternoon and evening, and has an impact on several aspects of sleep health, including increasing sleep onset latency, reducing TST and sleep efficiency, and prolonging REM sleep latency [25,26].

However, while the effects of acute caffeine intake are widely described, the effects of its daily consumption on sleep recovery following total sleep deprivation are not well documented or understood. Indeed, many laboratory studies are carried out after a withdrawal period [17,18,19,20,27]. In this way, accounting for daily habits of caffeine consumption has been suggested to be essential when analyzing the physiological effects [6,28]. We hypothesize that the habitual consumption of caffeine may interact with the effect of acute caffeine intake on recovery sleep following total sleep deprivation (TSD) (i.e., a 38 h period of continuous wakefulness).

The primary aim of our study was to evaluate the effects of acute caffeine intake on the characteristics of recovery sleep (i.e., macro- and micro-architecture and objective sleep quality) following TSD in healthy subjects without prior caffeine withdrawal. The secondary aim was to assess the interaction with daily caffeine consumption.

## 2. Materials and Methods

### 2.1. Participants

Forty-two participants aged 18 to 55 years were included in this study. The study followed the principles of the Helsinki Declaration (1975, revised in 2001) and the French medical laws and received approval from the CPP-Ile de France IV (Paris) Comité de protection des personnes (CPP) and has been declared at the Agence Nationale de la Sécurité du Médicament et des Produits de Santé (ANSM). The study has been registered in the clinical trial database (NCT03859882).

### 2.2. Inclusion and Exclusion Criteria

Participants were recruited through posters in our laboratories and at the Paris-Saclay University. They did not have active unresolved medical, psychiatric, or sleep disorders. These exclusion criteria were based on clinical interviews and subjective questionnaires: (I) the Hospital Anxiety and Depression Scale (HAD) (≥11) [29], (II) a significant medical history, (III) the Epworth Sleepiness Scale (ESS) (>10) [30], (IV) the Pittsburgh Sleep Quality Index (PSQI) (>8) [31].

We also excluded subjects with (V) an extreme chronotype assessed through the Morningness–Eveningness Questionnaire (score < 31 or >69) [32] or (VI) habitual time in bed per night <6 h. In addition, we excluded subjects (VII) habitually consuming more than 500 mg of caffeine per day in order to prevent the adverse events induced by withdrawal in the placebo condition [33]. In order to get as close as possible to the actual consumption habits of night-shift workers and to study the effect of daily habitual caffeine consumption, we did not exclude non-caffeine consumers.

The participants’ habitual caffeine consumption was assessed using a questionnaire [34,35] that included the following beverages and caffeine-containing foods: coffee with caffeine, tea, cola, other carbonated beverages with caffeine, and chocolate. For each item, participants were asked to indicate how often, on average, they had habitually consumed a given amount of each food or drink in the past year. Participants could choose from nine frequency categories (never, 1–3 per month, 1 per week, 2–4 per week, 5–6 per week, 1 per day, 2–3 per day, 4–5 per day, and 6 or more per day). Typical doses in milligrams [36] were assigned to each, and an approximate daily intake was obtained.

The participants were asked not to use medications with sleep-related side effects or illicit drugs or to abuse alcohol the week before the study. Participants did not travel between time zones within 7 days and did not work in shifts in the 2 weeks prior to the study. Participants were asked to complete a sleep–wake schedule during the week preceding the study and were asked to maintain their habitual daily caffeine consumption until they entered the laboratory protocol. In addition, participants were asked to maintain their habitual consumption of caffeine for two weeks prior to each laboratory period. Time in bed (TIB) during a control week between the two laboratory sessions was checked with actigraphy (Actiwatch TM, Cambridge Neurotechnology, Cambridgeshire, UK) to avoid starting the experiment with the subjects in sleep debt (i.e., sleep time under 6 h).

### 2.3. Sleep Deprivation Session

This study was conducted in the sleep laboratory of the Armed Forces Biomedical Research Institute (IRBA), Brétigny sur Orge, France. The ambient temperature was controlled and maintained at 22 ± 1 °C during all experiments. The brightness of the lighting was maintained between 150 and 200 lux during the awake periods, and the lights were turned off during sleep periods. Meals and caloric intake were standardized for all subjects (2600 kcal/day, including 20–30% proteins and 40–40% carbohydrates).

For each sleep deprivation session, participants remained inside the laboratory for 3 consecutive days. The experimental protocol included (I) a habituation/training day (D0), (II) a baseline day (D1) beginning at 07:00 until 00:00, (III) a total sleep deprivation (TSD) day beginning on D2 00:00 until 21:00 (i.e., 38 h of continuous wakefulness), and (IV) a recovery night until the end of the study (09:00 on D3) (Figure 1). Subjects were welcomed in groups of 4 at approximately 16:00 on D0.

Participants were not allowed to exercise or use tobacco, alcohol, or other psychoactive substances. When they were not engaged in testing, meals, or sleep periods, participants were allowed to read, to watch videos, or to speak with other participants or staff members and play games following a pre-established program. They were under visual surveillance by research staff members, and we used wrist actigraphy (Actiwatch TM, Cambridge Neurotechnology, Cambridgeshire, UK) to check that the subjects stayed awake during the 38-h continuous wakefulness period

### 2.4. Experimental Protocol

The study followed a double-blind, crossover, and placebo-controlled protocol, with two conditions: caffeine (CAF) and placebo (PBO). Each participant underwent two sessions of total sleep deprivation in random order of treatment (CAF vs. PBO). A 2-week washout period was applied between the two conditions, during which participants returned to their off-protocol lifestyle. For the caffeine condition, each participant received 2.5 mg per kg body weight of caffeine powder mixed in a decaffeinated beverage. We chose a moderate caffeine intake (~350 mg/day for a 70 kg adult), corresponding to the estimated average daily caffeine consumption in the French middle-aged working population (225 ± 161 mg/day) [37].

The placebo was the decaffeinated beverage. In order to design a placebo that could not be differentiated from the beverage containing the caffeine (smell, taste, color, bitterness), we used intensity 9 Nespresso ^®^ coffee (Nestlé SA, Vevey, Switzerland). The caffeine powder was added to intensity 3 Nespresso ^®^ coffee. We had previously verified that 6 blinded members of the staff (3 successive tests) were unable to distinguish the drink containing caffeine from the placebo (10 errors out of 18 tests). We also checked that the decaffeinated drink did not contain caffeine. The caffeine powders had been pre-measured by the project supervisor.

The beverage was administered at 08:30 and 14:30 each day during total sleep deprivation (i.e., after 1.5, 7.5, 25.5, and 31.5 h of prolonged wakefulness, 6.5 h before bedtime for the last one). The caffeine powder was pre-measured by the project supervisor. This amount of caffeine powder was chosen for its attention-enhancing properties in sleep-deprived conditions (2.5–8 mg/kg of caffeine) [38]. Caffeine or placebo was administered in a decaffeinated beverage.

### 2.5. Sleep Recording and Analysis

Polysomnographic recordings were performed using a connected headband (Dreem^®^, Paris, France) during the nights before (baseline) and after sleep deprivation (recovery night). The headband is a validated wireless recording device [39] including 5 dry nano-carbon EEG electrodes (Fpz, Fp1, Fp2, M1, and M2) in order to create 4 EEG channels (Fp1-M1, Pf2-M2, Fp1-Fpz, Fp1-Fp2). EEG signals were sampled at 250 Hz and filtered between 0.4 and 18 Hz. The headband also includes an accelerometer and an SpO_2_ probe. The headband was used without auditive activity.

The participants were asked to wear the connected headband from 21:00 to 07:00. A member of the staff checked the headband position and the signal quality through the dream software. Each participant could sleep ad libitum between 21:00 and 07:00. In order to reduce the risk of discomfort caused by the headband and to respect habituation, the headbands were given to the subjects 6 days before the study so that they could wear them at home.

Sleep stages were automatically analyzed using the Dreem software [39], and the hypnograms were extracted. Raw EEG signals and hypnograms were computed and analyzed using the lunaR library for the R statistical package (http://zzz.bwh.harvard.edu/luna/ (V1.00, 31 May 2024, accessed on 1 June 2024)) for the determination of sleep microstructure parameters.

The classical macrostructure sleep variables considered in this study were time in bed (TIB; i.e., time from lights off to final awakening, in minutes), actual sleep time (AST, time from the first appearance of N1 to the last sleep epoch, in minutes), total sleep time (TST; i.e., time spent in NREM and REM sleep stages, in minutes), sleep onset latency in minutes (SOL), sleep stages’ durations (N1, N2, N3, REM) and proportions to TST, the duration of wake after sleep onset over TST (WASO), and sleep efficiency (SEI%; i.e., percentage of TST over TIB).

Moreover, objective sleep quality was evaluated through a set of variables [40]:Sleep continuity/fragmentation: total awakening frequency per hour of AST; frequency of brief (<2 min) and long (>2 min) awakenings per hour of AST; frequency of awakenings from N1, N2, N3, and REM per minute of that stage.Sleep stability: number and average duration of N1, N2, N3, and REM sleep periods not interrupted by a wake period longer than 2 min, frequency of stage transitions (stage transitions were defined as all transitions from one stage to another, including all those to and from wakefulness) calculated per minute of each stage.Sleep organization: number of complete sleep cycles, defined as the sequences of NREM and REM sleep (each lasting at least 10 min) not interrupted by a wake period longer than 2 min, and the mean duration of cycles in minutes and % AST [40,41].Sleep intensity: EEG power spectral analysis was performed for each EEG frequency band: slow oscillation (0.5–1 Hz), delta (1–4 Hz), theta (4–8 Hz), alpha (8–12 Hz), sigma (12–15 Hz), and beta (15–20 Hz). For the present analysis, data derived from the frontal EEG electrode (i.e., Fp1-M1) in NREM stages were used. Artifacts due to electrocardiogram interference were removed using a template subtraction method. Manual visual adjudication was performed by a researcher who was blinded to the subject group, and spectral data with significant artifacts were excluded manually. In accordance with Welch’s method, the spectral power density was calculated using 10 overlapping 4 s sub-epochs for each 30 s epoch, with a 50% tapered cosine window [6].

### 2.6. Statistical Analysis

The statistical analyses were performed using Jamovi (version 1.6.15, 2022) for R. Values are expressed as the mean ± SEM. Sleep parameter values were analyzed using a linear mixed model including two fixed effects for night (repeated measures, after sleep deprivation vs. before [i.e., recovery night vs. baseline night]) and acute treatment (caffeine vs. placebo, repeated measure) and a continuous effect (covariable) for daily habitual caffeine consumption (non-repeated measure). Subjects were included in the model as random effects for night, treatment, and night × treatment interaction. The Satterthwaite method was used for the degrees-of-freedom determination. Effect sizes were estimated with the calculation of the eta square (η^2^ > 0.01 indicates a small effect, η^2^ > 0.06 a medium effect, and η^2^ > 0.14 a large effect). In the case of a significant main effect or interaction, significant differences between conditions were identified using Tukey post hoc tests. In order to manage multiple comparison biases, we used the Benjamini and Hochberg method for the type 1 error correction [42]. Corrected *p* values < 0.05 were considered to be significant. Correlations between parameters were made using the Pearson coefficient.

## 3. Results

There were 42 participants in this study. One participant was excluded due to a serious adverse reaction to caffeine (vomiting, severe nausea). Finally, a total of 41 healthy participants (33.2 ± 0.9 years) followed the protocol, including 22 females and 19 males. Participants’ average daily caffeine consumption was 247 ± 23 mg (17% never consumed caffeine). The mean body mass index (BMI) was 22.7 ± 0.6 kg·m^2^ for men and 23.3 ± 0.7 kg·m^2^ for women. The mean weekly exercise duration was 3.1 ± 0.4 h. The mean nightly TST was 7.2 ± 0.2 h, and the average sleepiness score (ESS) was 6.8 ± 0.6. Chronotypes were distributed as follows: 30% of subjects were of the intermediate chronotype, 35% were of the morning chronotype, and 35% were of the evening chronotype.

### 3.1. Sleep Macrostructure Parameters

We observed an increase in total sleep time (TST) during the recovery night (Figure 2) compared to the baseline night before sleep deprivation (+110.2 ± 23.2 min) (F_(1,36)_ = 197.6, *p* = 0.001, η^2^ = 0.21), with an interaction with habitual caffeine consumption (F_(1,95)_ = 3.94, *p* = 0.04, η^2^ = 0.08). Acute caffeine intake decreased TST during the recovery night (F_(1,70)_ = 6.31, *p* = 0.01, −30.2 ± 8.2 min, η^2^ = 0.11) (see Appendix A, Supplementary Data for linear mixed model statistical tables).

N1 duration decreased during the recovery night (F_(1,93)_ = 9.68, *p* = 0.002, η^2^ = 0.18), without an interaction with acute or chronic caffeine consumption (or treatment). We did not observe changes in N2 sleep duration, but we observed a decrease in the N2 proportion (F_(1,97.6)_, *p* < 0.003). No effect of acute caffeine treatment or chronic consumption was observed. The duration and proportion of N3 sleep increased during the recovery night (F_(1,93)_ = 30.28, *p* = 0.002, η^2^ = 0.18 and F_(1,93)_ = 46.36, *p* = 0.001, η^2^ = 0.19), with a significant effect of caffeine treatment on recovery sleep (interaction, F_(1,93)_ = 4.22, *p* = 0.04, η^2^ = 0.06, 17% decrease). There was no interaction with habitual caffeine consumption (see Appendix A).

Concerning sleep latencies, we observed during the recovery night, in comparison to the baseline, significant decreases in sleep onset latency (SOL) (F_(1,103.8)_ = 11.83, *p* < 0.001, η^2^ = 0.19) and N3 sleep stage latency (F_(1,104)_ = 57.96, *p* = 0.001, η^2^ = 0.14) and an increase in REM sleep latency (F_(1,104)_ = 5.91, *p* = 0.02, η^2^ = 0.12). There was no interaction with caffeine treatment or habitual caffeine consumption in sleep latency (see Appendix A).

We also observed an increase in the number of WASO periods (interaction treatment × night, F_(1,104)_ = 5.31, *p* = 0.04; +61.01 ± 12.3%) in the caffeine condition. (see Appendix A).

### 3.2. Sleep Continuity/Fragmentation, Stability, and Organization and EEG Power Density

We observed an interaction between night and treatment on the number of N1 and N3 sleep stages (F_(1,103)_ = 9.8, *p* = 0.02, η^2^ = 0.12 and F_(1,103)_ = 5.3, *p* = 0.04, η^2^ = 0.06), with higher numbers in the caffeine compared to placebo condition (*p* = 0.03, Figure 3). The average N3 stage duration increased during recovery sleep in comparison to the baseline night, and the N3 and REM sleep stage durations were lower during the recovery night in the caffeine compared to the placebo condition. We observed an increase in the transition rate (F_(1,93)_ = 9.3, *p* = 0.03, η^2^ = 0.09) during the recovery night in comparison to the baseline night in the caffeine condition and a higher rate value in the caffeine compared to the placebo condition.

In the placebo condition, the absolute power density of slow and delta oscillations increased (F_(1,93)_ = 6.8, *p* = 0.03, η^2^ = 0.09; F_(1,93)_ = 11.4, *p* = 0.02, η^2^ = 0.12) during the recovery night in comparison to the baseline night, and a decreasing effect of caffeine intake was observed during the recovery night (*p* = 0.03) (Figure 3). No effect of habitual caffeine consumption was observed.

No difference between conditions was observed for the number of brief awakenings. However, we observed less sleep continuity during the recovery night in the caffeine condition than in the placebo condition, with a higher frequency of long awakenings during the recovery night in comparison to the baseline night (*p* = 0.03) and the placebo recovery night (*p* = 0.02) (interaction, F_(1,104)_ = 6.72, *p* = 0.03, η^2^ = 0.10). In the placebo condition, we observed, during the recovery night, a decreased awakening frequency from N1 and N2 in comparison to the baseline night. A larger increase in awakening frequency was observed from the N1 and N2 stages. No interaction with habitual caffeine consumption was observed.

Concerning sleep stability (Table 1), we observed a night effect, with decreased arousal frequencies from N2 to N1 (F_(1,104)_ = 4.53, *p* = 0.04, η^2^ = 0.08) and N3 to N1 cycles (F_(1,104)_ = 5.23, *p* = 0.03, η^2^ = 0.11) during the recovery night compared with the baseline night in the placebo condition. During the recovery night, the caffeine condition was associated with a higher transition frequency from N3 to N1 cycles compared to the placebo condition (F_(1,104)_ = 4.72, *p* = 0.04, η^2^ = 0.08). No effect of caffeine treatment or night was observed for the other stages’ transitions. We did not observe an interaction with habitual caffeine consumption.

Sleep organization changed during the recovery night compared to the baseline night, with a higher number of cycles (F_(1,104)_ = 6.53, *p* = 0.03, η^2^ = 0.10) and a longer time spent in a cycle in % AST (F_(1,104)_ = 10.33, *p* = 0.01, η^2^ = 0.14) in the placebo condition. Caffeine intake decreased the relative time in a sleep cycle (F_(1,104)_ = 6.12, *p* = 0.03, η^2^ = 0.09). No interaction with habitual caffeine consumption was observed.

### 3.3. Correlations between Sleep Parameters and Daily Habitual Caffeine Consumption

We observed a negative correlation (Figure 4) between daily habitual caffeine consumption and TST only during the recovery night in the caffeine (R^2^ = 0.32, *p* = 0.01) and placebo (R^2^ = 0.25, *p* = 0.02) conditions. No significant correlation between TST and habitual caffeine conditions was observed during the baseline night. No effect of daily habitual consumption was observed for the REM, N2, or N1 sleep stage duration. For the duration of WASO, we observed, during the recovery night, a positive correlation with daily caffeine consumption (R^2^ = 0.41, *p* = 0.001) in the caffeine condition only. The frequency of long awakenings was also positively correlated with daily caffeine consumption (R^2^ = 0.34, *p* < 0.01), without an interaction with the day or treatment.

## 4. Discussion

In the present study, we show that a moderate caffeine intake during a total sleep deprivation period alters sleep during the subsequent recovery night, with decreases in total sleep time (TST) (i.e., less sleep rebound), N3 sleep duration, and sleep quality, as assessed through sleep continuity/fragmentation, stability, organization, and EEG delta band density. For the first time, we show that the daily habitual consumption of caffeine is negatively correlated with TST during the recovery night but not during the baseline night in both caffeine and placebo conditions (low TST in higher consumers). It also positively correlated with the WASO duration and the frequency of long awakenings during recovery sleep in the caffeine condition: the higher the daily consumption, the longer the WASO duration and the higher the frequency of long awakenings.

In order to respect the participants’ chronotypes and sleep habits and not limit the recovery sleep period, the sleep opportunity was between 21:00 and 07:00. Under these conditions, we observed similar or higher TST values (>500 min for both caffeine and placebo groups) during the recovery night compared to those in the literature, where the time spent in bed (TIB) is adapted to the subjects’ habitual time in the sleep–wake cycle (e.g., around 23:00 and TST < 470 min for both caffeine and placebo groups) [18] or set to 8 h’ time in bed [43].

Few studies have evaluated, in humans, the effects of acute caffeine on nighttime sleep recovery following one night of sleep deprivation [18,20], and several have explored the additional question of daytime sleep recovery or daytime sleepiness [20,22,23,44]. During a 5 h episode of recovery daytime sleep following 23 h of being awake, prior caffeine intake (200 mg approximately 6 h before sleep) significantly reduced the SWS percentage and sleep efficiency, as well as SWS and TST minutes, with significant increases in WASO, SOL, and the duration and number of awakenings compared to the placebo condition [23].

Similar results were evidenced by Carrier et al. [22], as caffeine intake (100 mg of 3 h before bedtime following 25h of sleep deprivation, and the remaining dose 1 h before bedtime) decreased SWS and increased sleep latency during daytime sleep recovery compared with placebo.

In contrast, 4 mg/kg of caffeine administered on three consecutive nights, in two equal doses (at 0220 and 0120 h) each night, decreases physiological sleepiness during nighttime hours and, at least initially, enhances performance, with no significant consequences on the polysomnographic measurement of daytime sleep [20,44].

Differences in the results of recovery sleep characteristics following sleep deprivation and caffeine intake may be explained by differences in the choice of recovery sleep type (nighttime or daytime sleep), the duration of sleep deprivation, the amount of caffeine consumed, the time of last caffeine intake, or habitual caffeine consumption. For example, Beaumont et al. [20] observed no significant effect of slow-release caffeine (300 mg/dose) on nighttime recovery sleep after a more prolonged sleep deprivation period than in our study (i.e., 64 h of wakefulness) [20]. In this study, the last dose of slow-release caffeine was 14h before recovery sleep. These authors only observed an increase in SWS during the first recovery night and REM sleep during the second night in the two groups (placebo and caffeine), which are markers of elevated homeostatic sleep pressure [20]. In order to get as close as possible to the actual consumption habits of night workers and to study the effect of daily habitual caffeine consumption, we did not exclude non-caffeine consumers. This was possible given the relatively large number (n = 41) of subjects in our work for a total sleep deprivation protocol.

This point is important in the context of increasing caffeine intake in the afternoon and early evening and the increasingly popular use of caffeinated energy drinks, as well as the higher caffeine content of premium coffee [24]. We excluded from our work subjects usually consuming more than 500 mg of caffeine per day in order to prevent withdrawal-induced adverse effects in the placebo condition [33]. Moreover, using more than 500 mg of caffeine is associated with side effects and high levels of sleep disorders [33,45]. In our experimental protocol, the last caffeine intake took place around 6 h before bedtime following 38 h of wakefulness and had an impact on nighttime recovery sleep. In the condition of a regular sleep–wake cycle, the literature indicates that caffeine should be consumed at least 8 to 8.8 h prior to bedtime to avoid a decrease in total sleep time [15]. We show in our work that a moderate dose of caffeine (i.e., ~350 mg/day), 6 h prior to bedtime, is associated with reductions in total sleep time, total N3 sleep stage duration, and the durations of N3 periods and REM periods.

In addition, we observed a decreased EEG delta power density. The caffeine condition increased the rate of sleep stage transitions and decreased markers of sleep quality. This acute caffeine effect on the nighttime sleep recovery macrostructure following 40 h of wakefulness was not described by Landolt et al. [18], as they found that total sleep time, sleep efficiency, and slow-wave sleep were increased similarly in their caffeine and placebo groups. The only result in their study consistent with our findings is the reduction in EEG power in the 0.75–2.0 Hz band. In the same way, an acute caffeine effect on the nighttime sleep macrostructure was not found by Beaumont et al. [20] following 64 h of continuous wakefulness. In these two previous studies, the absence of acute caffeine effects on nighttime recovery sleep following prolonged wakefulness was likely due to the timing of the last caffeine intake, i.e., 14 h before bedtime. Our results are closer to what has been described for daytime sleep recovery following 25 or 28 h of sleep deprivation with caffeine taken 6 h before bedtime [22,23].

The reducing effect of acute caffeine on the slow-wave sleep (SWS) duration and EEG delta power density observed in our study during nighttime sleep recovery may have implications for cognitive functions [46]. We did not find studies related to cognitive responses the day after nighttime recovery sleep following one night of total sleep deprivation (equivalent to 38 h of continuous wakefulness). Even in our previous results, we showed beneficial effects of acute caffeine on sustained attention during total sleep deprivation until 32 h of wakefulness, but we did not evaluate it the day after the recovery night [47]. The Beaumont et al. [20] study evidenced that cognitive performance was objectively impaired the following day after 64 h of continuous wakefulness, but without a difference between the slow-release caffeine and placebo groups either in this recovery performance or in sleep macrostructure the night before. To our knowledge, the only study that showed an acute caffeine effect on nighttime recovery associated with subsequent daytime cognitive performance concerned chronic sleep restriction [48]. In this study, in healthy good sleepers, acute caffeine intake was demonstrated to negatively impact the subsequent recovery of cognitive processes after 5 days of moderate sleep restriction (5 h time in bed per night) because the caffeine group was slower to return to baseline in sustained attention in the PVT task during the recovery period compared to the placebo group [48]. In association with this, the N3 duration in the caffeine group was higher than in the placebo group on the second recovery night only, and the authors indicated that caffeine created a greater homeostatic sleep need. However, the paradox associated with a longer N3 sleep duration and degraded sustained attention in the caffeine condition in this study could be related to the fact that the second night of recovery followed a day without caffeine intake [48].

Considering other sleep indexes, we show, for the first time, the impact of acute caffeine intake on nighttime sleep recovery (i.e., following total sleep deprivation), stability (e.g., arousal and state transition frequencies), continuity/fragmentation (awakening frequency), and organization (e.g., number of sleep cycles and time spent in cycles), which have been proposed to objectively determine sleep quality and to potentially play a role in memory consolidation, based on many experimental results [40,41,49]. In this sense, it has been recently proposed that the “recovery” function of sleep relies on the NREM-REM alternation rather than merely on slow-wave activity [50]. Our results showed that acute caffeine increased the number of N1 and N3 sleep stage periods but decreased the durations of N3 and REM sleep periods during recovery sleep in the caffeine compared with the placebo condition.

Caffeine also affects sleep continuity/fragmentation (i.e., increased number of long awakenings per hour of actual sleep time and increased frequency of awakenings from N1 sleep), stability (i.e., increased stage transition frequency), and organization (i.e., decreased mean cycle duration per hour of actual sleep time).

In our study, we confirmed in the placebo condition that, during nighttime recovery sleep following one night of sleep deprivation, the durations of TST, N3, and REM and the sleep efficiency index were higher, while the N1 duration and sleep latency were lower, compared to baseline sleep [4,18,51,52].

We also confirmed that the night of sleep recovery increases power in the delta frequency in NREM sleep in the placebo condition [18]. The spectral delta wave power depends not only on the amount of sleep time spent in deep slow-wave sleep but also on the amplitude of slow waves, making it more of a marker of the intensity of this deep slow-wave sleep. Indeed, its intensity in recovery sleep following total sleep deprivation and baseline sleep was associated with the restoration of vigilance in the PVT task [4], but its increase was found to predict a decrease in vigilance and inhibition control after nights of moderate sleep restriction [53]. These findings suggest that EEG signatures may partially explain cognitive impairment caused by sleep loss.

To conclude, we described acute caffeine effects on nighttime sleep recovery following 38 h of wakefulness on sleep macro- and micro-structure and sleep continuity, stability, and organization, highlighting the role of sleep continuity and stability in effective sleep-dependent consolidation processes [54,55]. The interest in adding sleep indicators of quality, such as continuity, stability, and organization, has been previously indicated, as bad sleepers and insomniacs presented impairments in sleep macrostructure (sleep latency, sleep efficiency, WASO) and in continuity, stability, and organization, whereas poor sleepers showed disrupted continuity and stability [40,56]. Interestingly, a longer WASO was found to be associated with poorer cognitive performance (sustained attention and inhibition) in a sleep restriction protocol [53].

With respect to the influence of daily caffeine consumption on the sleep-related effects of acute caffeine intake in adults, there are scarce data. In 2015, it was shown that caffeine consumption in adolescents may lead to later bedtimes and reduced slow-wave activity (SWA), a well-established marker of sleep depth [57]. A recent study showed that in young adult regular caffeine consumers (478.1 ± 102.8 mg/day), who can be considered high consumers, acute caffeine intake for 10 days (3 × 150 mg per day) does not affect total sleep time or sleep architecture, while it increases REM sleep latency [25]. We also demonstrated that high daily habitual caffeine consumption decreases attentional performance and EEG alpha frequencies, decreasing tolerance to sleep deprivation [8]. In the study herein, daily caffeine consumption was negatively correlated with total sleep time on the night of recovery following total sleep deprivation in the caffeine and placebo conditions and positively correlated with the WASO duration and the frequency of long awakenings in the caffeine condition only, likely due to the fact that chronic caffeine use impacts the adenosine receptor system [58,59].

One limitation of our work is that we did not assess performance after the recovery night. Future studies taking into account performance after the recovery night following total sleep deprivation will enable us to identify the main sleep parameters associated with sleep efficiency and quality. Another limitation is the small number of participants used to investigate the link between daily caffeine consumption and sleep characteristics. Further studies with a different design could better assess this association, particularly by including higher caffeine consumers.

## 5. Conclusions

Our results originally show that a moderate acute caffeine intake (~350 mg/day) 6.5 h before bedtime following 38 h of continuous wakefulness reduced total sleep time and the N3 sleep stage. It also altered recovery sleep continuity (increased number of long awakenings), stability (higher stage transition frequency), and organization (shorter time spent in a complete sleep cycle) and decreased the delta power spectral density during NREM sleep. Correlation analysis showed that daily caffeine consumption interacted with the acute effects of caffeine on recovery sleep, and the higher the daily consumption, the longer the WASO duration and the higher the frequency of long awakenings. Our results could be interesting for individuals working night shifts and consuming caffeine. To avoid reducing nighttime sleep duration, coffee should be consumed at least 8.8 h before bedtime, as already recommended in the review of Gardiner et al. [15], and a reduction in daily coffee/caffeine consumption should be recommended. To maintain appropriate alertness during a prolonged period of wakefulness, a low dose of caffeine would be advisable [47], in addition to paying attention to the time of the last intake before the recovery night and also taking care to maintain a habitual daily consumption of less than 3 espressos per day.

## Figures and Tables

**Figure 1 nutrients-16-03442-f001:**
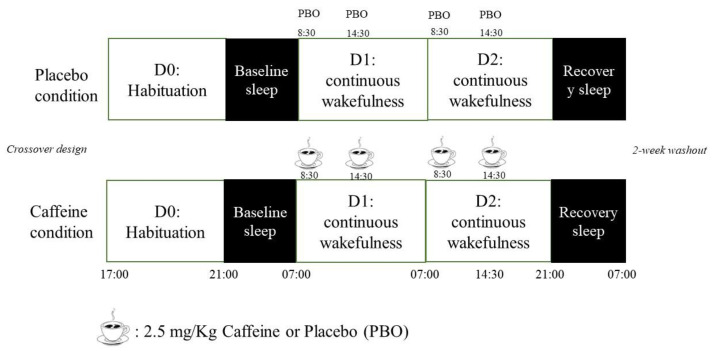
Study design.

**Figure 2 nutrients-16-03442-f002:**
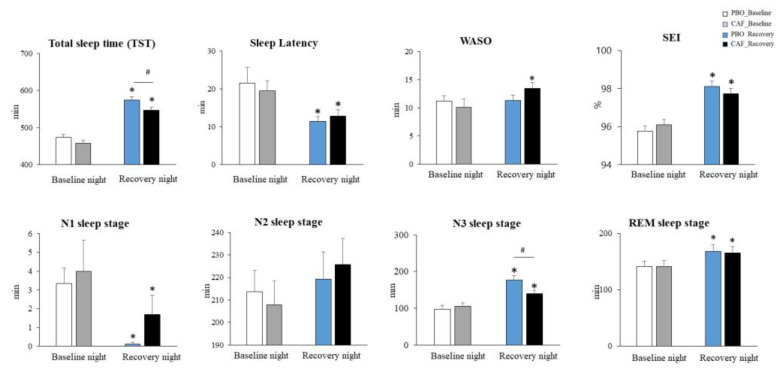
Changes in sleep parameters in caffeine (CAF) and placebo (PBO) conditions. * Indicates a difference between the night of sleep recovery following total sleep deprivation and the baseline night before total sleep deprivation; # indicates a difference between caffeine (CAF) and placebo (PBO) conditions.

**Figure 3 nutrients-16-03442-f003:**
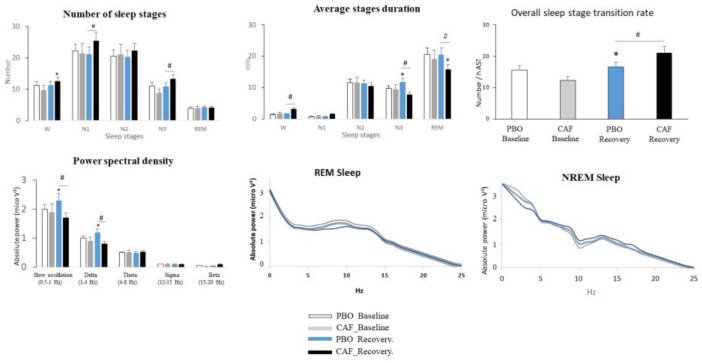
Changes in sleep microstructure parameters: stability, continuity, and density. * Indicates a difference between the night of sleep recovery following total sleep deprivation and the baseline night before total sleep deprivation; # indicates a difference between caffeine (CAF) and placebo (PBO) conditions (*p* < 0.05).

**Figure 4 nutrients-16-03442-f004:**
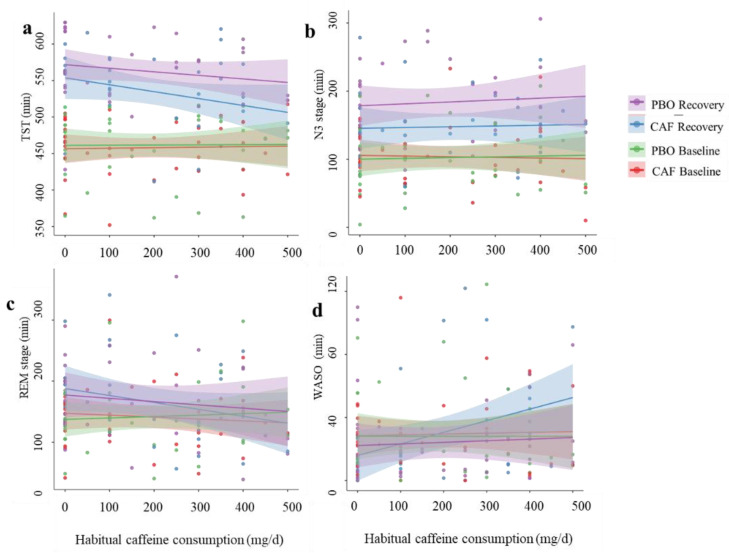
Correlations (with confidence intervals) between habitual caffeine consumption and sleep parameters. (**a**) Total sleep time, (**b**) N3 sleep stage, (**c**) REM sleep, and (**d**) wake after sleep onset (WASO) in the four conditions. CAF is caffeine, and PBO is placebo.

**Table 1 nutrients-16-03442-t001:** Sleep continuity/fragmentation, stability, and organization in the four conditions.

	PBO Baseline	CAF Baseline	PBO Recovery	CAF Recovery
Sleep continuity				
Awakening (/h·AST)				
Brief (<2 min)	3.08 ± 1.23	3.12 ± 1.22	3.21 ± 1.26	4.13 ± 2.22
Long (>2 min)	0.18 ± 0.11	0.20 ± 0.18	0.16 ± 0.20	0.82 ± 0.20 *^#^
Awakening frequency ^a^				
From N1 (/min)	0.12 ± 0.07	0.13 ± 0.05	0.05 ± 0.06 *	0.13 ± 0.06 ^#^
From N2 (/min)	0.41 ± 0.03	0.41 ± 0.02	0.36 ± 0.02 *	0.48 ± 0.02 ^#^
From N3 (/min)	0.03 ± 0.03	0.03 ± 0.04	0.02 ± 0.06	0.04 ± 0.05
From REM (/min)	0.04 ± 0.03	0.03 ± 0.06	0.03 ± 0.02	0.03 ± 0.02
Sleep stability, Arousal frequency ^a^				
N2 to N1 (/min)	0.13 ± 0.07	0.11 ± 0.04	0.08 ± 0.08 *	0.11 ± 0.08
N3 to N2 (/min)	0.39 ± 0.23	0.27 ± 0.22	0.25 ± 0.22	0.31 ± 0.28
N3 to N1 (/min)	0.03 ± 0.05	0.03 ± 0.04	0.01 ± 0.04 *	0.03 ± 0.03
REM to N1 (/min)	0.02 ± 0.01	0.02 ± 0.03	0.04 ± 0.01	0.03 ± 0.02
Sleep organization				
Complete sleep cycles (n)	0.91 ± 0.81	1.12 ± 1.01	2.21 ± 1.71 *	1.22 ± 1.21
Cycle duration (min)	30.11 ± 22.61	32.61 ± 21.10	42.41 ± 26.11	32.21 ± 20.11
Cycle duration (%AST)	12.21 ± 14.11	12.54 ± 15.11	31.12 ± 20.61 *	13.22 ± 19.11 ^#^

^a^ Awakening and arousal frequencies are calculated as frequencies over the total time spent in each stage in minutes. * Indicates a difference between the night of sleep recovery following total sleep deprivation and the baseline night before total sleep deprivation; ^#^ indicates a difference between caffeine (CAF) and placebo (PBO) conditions (*p* < 0.05).

## Data Availability

Data are available by request to the corresponding author.

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
