# Peer review of "Caffeine Intake Alters Recovery Sleep after Sleep Deprivation"

_nutrients, 2024, doi:10.3390/nu16203442_

Round 1

Reviewer 1 Report

Comments and Suggestions for Authors

The authors analyzed the effect of caffeine consumption during the period of continuous wakefulness on recovery sleep. The study was very carefully planned and conducted. The appropriate parameters were used to assess sleep quality, defining macro- and microstructure as well as continuity, stability and organization of sleep. The analysis took into account not only acute caffeine consumption, but also daily consumption typical for individual subjects. I have no reservations about the purpose, scope and method of conducting the research and analysis. However, in the conclusions the authors postulate that in order to avoid a negative effect on sleep, the last portion of caffeine (coffee) should be consumed at least 8 hours before sleep. This is not a postulate resulting from the conducted research, because according to the experimental plan, the last portion of caffeine was consumed 6.5 hours before sleep. The authors also outline the project of future research. It seems to me that it would also be worthwhile to determine a dose of caffeine sufficient to maintain appropriate alertness during sleep deprivation, and not significantly disrupting recovery sleep.

Author Response

We thank the reviewer for the response. Regarding the remark in our conclusion, we added the Gardiner et al. (2023) reference, and stuck to the advice given by their review of a final intake of caffeine (coffee) that should be consumed at least 8.8 hours before sleep.

As proposed, we added in conclusion that taking low dose caffeine prolonged wakefulness would be advisable to maintain appropriate alertness, but paying attention to the time of the last intake before the night of recovery, and also taking care to maintain a daily habitual daily consumption of less than 3 espressos per day.

Reviewer 2 Report

Comments and Suggestions for Authors

Thank you for the opportunity to review the manuscript of Pauchon et al aiming to assess the effect of acute caffeine intake on the duration and quality of recovery sleep following total sleep deprivation. I congratulate the authors for choosing such an interesting subject and the quality of the research. The Introduction is well written and nicely introduce the research objective. The results are clearly presented and discussed in the context of existing literature and the conclusions are based on the study results.

The methodology seems appropriate, but would benefit of additional details:

1.     Please briefly describe the placebo used to match caffeine, as you state that the study was double-blind. If the same decaffeinated beverage was used as in the caffeine condition, how was masked the taste of the caffeine powder?

2.     There is evidence that the acute effect of caffeine in users and non-users may be different, including the effect on brain activity. The inclusion of caffeine non-users may have influenced the final results. Please explain in the Methodology why caffeine non-users were included or discuss this in the Discussion section.

Below are also listed few minor issues:

1. Abbreviations should be defined at their first use. Thereafter please use only the abbreviation thought the main text of the manuscript. For example “slow-wave sleep (SWS)” on page 2, “slow-wave sleep” on page 10, “slow wave sleep (SWS)” on page 10, “slow-wave sleep” on page 11.

2. On page 10 – the text starts with 2 bullet points that were probably forgotten.

3. On page 10 – please replace “(Gardiner 2023)” with reference [15].

Comments on the Quality of English Language

No comments.

Author Response

We thank the reviewer for their remarks.

The methodology seems appropriate, but would benefit of additional details:

1.Please briefly describe the placebo used to match caffeine, as you state that the study was double-blind. If the same decaffeinated beverage was used as in the caffeine condition, how was masked the taste of the caffeine powder?

The placebo has now been described as follows “The placebo was the decaffeinated beverage. In order to design a placebo that could not be differentiated from the beverage containing the caffeine (smell, taste, color, bitterness), we used an intensity 9 Nespresso ® coffee (Nestlé SA, Vevey, Switzerland). The caffeine powder was added to an intensity 3 Nespresso ® coffee. We have previously verified that 6 blinded members of the staff (3 successive tests) were unable to distinguish the drink containing caffeine from the placebo (10 errors out of 18 tests). We also checked that the decaffeinated drink did not contain caffeine. The caffeine powders have been pre-measured by the project supervisor.”

2.There is evidence that the acute effect of caffeine in users and non-users may be different, including the effect on brain activity. The inclusion of caffeine non-users may have influenced the final results. Please explain in the Methodology why caffeine non-users were included or discuss this in the Discussion section.

We did not exclude non-caffeine users, as we chose to keep as close as possible to real-life conditions, given the large number of subjects for a sleep deprivation study (n=41). We preferred to exclude subjects consuming more than 500 mg of caffeine daily (i.e. more than 7 espressos) because of the side-effects and physiological stress of total sleep deprivation. A dedicated paragraph is included in the discussion and we added a sentence in the methodological paragraph.

Below are also listed few minor issues:

Abbreviations should be defined at their first use. Thereafter please use only the abbreviation thought the main text of the manuscript. For example “slow-wave sleep (SWS)” on page 2, “slow-wave sleep” on page 10, “slow wave sleep (SWS)” on page 10, “slow-wave sleep” on page 11.

We apologize and have now corrected to “slow-wave sleep”.

On page 10 – the text starts with 2 bullet points that were probably forgotten.

They have been deleted.

On page 10 – please replace “(Gardiner 2023)” with reference [15].

We apologize and have now corrected this point.

Reviewer 3 Report

Comments and Suggestions for Authors

The authors have conducted a research study on the effects of caffeine consumption on recovery sleep after a determined period of sleep deprivation. It is well-structured and comprehensive, providing insights into changes in sleep parameters following caffeine consumption and investigating the effects of habitual caffeine consumption on the results. The topic is relevant considering the known side effects of inadequate sleep, especially in shift workers.

The introduction effectively establishes the study's context and significance. It also provides a concise overview of the current literature on the subject and clearly outlines the study's objectives.

The study methodology is presented in great detail and is easily reproducible. However, some areas require adjustments. No information about the participants' recruitment process is provided. I recommend mentioning this. Also, socioeconomic status or general health habits are factors that can affect the results and might be considered.  The dose of caffeine administrated is 2,5/kg. Can you provide more details about the rationale behind choosing this exact dose? The authors should consider that the sensitivity to caffeine differs between individuals and that different metabolisms might influence the results. Also, I suggest that attention be paid to the control of environmental factors. Standardized calorie intake is provided, but there isn’t any information about the timing of meals or the macronutrients in each meal that might influence sleep quality. 

The results are comprehensive and well-organized, and the tables are clearly explained.

I suggest adding a paragraph with the study’s limitations to increase transparency. Although some limits are provided, such as not assessing participants' performance after the recovery night, other limitations, such as the small sample size or excluding heavy coffee drinkers and their effect on generalization of results, should be specified and explained.

The references are well-chosen, with the majority published in the last five years.

Author Response

The authors have conducted a research study on the effects of caffeine consumption on recovery sleep after a determined period of sleep deprivation. It is well-structured and comprehensive, providing insights into changes in sleep parameters following caffeine consumption and investigating the effects of habitual caffeine consumption on the results. The topic is relevant considering the known side effects of inadequate sleep, especially in shift workers.

The introduction effectively establishes the study's context and significance. It also provides a concise overview of the current literature on the subject and clearly outlines the study's objectives.

The study methodology is presented in great detail and is easily reproducible. However, some areas require adjustments. No information about the participants' recruitment process is provided. I recommend mentioning this. Also, socioeconomic status or general health habits are factors that can affect the results and might be considered.  The dose of caffeine administrated is 2,5/kg. Can you provide more details about the rationale behind choosing this exact dose? The authors should consider that the sensitivity to caffeine differs between individuals and that different metabolisms might influence the results. Also, I suggest that attention be paid to the control of environmental factors. Standardized calorie intake is provided, but there isn’t any information about the timing of meals or the macronutrients in each meal that might influence sleep quality. 

            We thank the reviewer for his/her help. We provide information about the recruitment. We give some information about how meal have been balanced between protein (20-30%), carbohydrate (40-50%) and fats (20-30%)..

We provide more details about the rational behind the chose of the dose of caffeine administered. In our study, we assessed a moderate caffeine intake (~350 mg/day for a 70 kg adult), corresponding to the estimated average daily caffeine consumption in a French middle-aged working population (225 ± 161 mg/day)

The results are comprehensive and well-organized, and the tables are clearly explained.

I suggest adding a paragraph with the study’s limitations to increase transparency. Although some limits are provided, such as not assessing participants' performance after the recovery night, other limitations, such as the small sample size or excluding heavy coffee drinkers and their effect on generalization of results, should be specified and explained.

We added a paragraph at the end of the manuscript

The references are well-chosen, with the majority published in the last five years.

Reviewer 4 Report

Comments and Suggestions for Authors

 appreciate the opportunity to review your manuscript and commend your efforts on this study. Your work offers valuable insights, yet I have several suggestions for improvement that would enhance the clarity and overall rigor of the paper.

  1. I suggest replacing the term subjects with participants throughout the entire manuscript, as it aligns better with contemporary research standards and ethical guidelines.

  2. In the Subjects section, it would be beneficial to include detailed characteristics of the enrolled sample. Additionally, please explain how the enrollment process was applied. Clarifying these points will provide readers with a better understanding of your study's scope and sample representativeness.

  3. I recommend moving the inclusion and exclusion criteria into a separate subsection under the methods section. This will improve the organization of the methods and make it easier for readers to follow your experimental design.

  4. The reporting of results is somewhat unclear in its current form. For each result, please specify the type of data analysis conducted and ensure that all relevant statistics are provided. This should include effect sizes, which are currently reported only for some analyses. Including consistent effect size reporting across all analyses will help to provide a clearer interpretation of the results.

  5. Finally, please include a Limitations subsection within the discussion. This will allow you to acknowledge any potential weaknesses in the study, such as sample size limitations or methodological constraints, and demonstrate transparency in the interpretation of the findings.

Author Response

We appreciate the opportunity to review your manuscript and commend your efforts on this study. Your work offers valuable insights, yet I have several suggestions for improvement that would enhance the clarity and overall rigor of the paper.

I suggest replacing the term subjects with participants throughout the entire manuscript, as it aligns better with contemporary research standards and ethical guidelines.

We thank the reviewer for her/his remark and help to improve our manuscript. We have corrected accordingly.

In the Subjects section, it would be beneficial to include detailed characteristics of the enrolled sample. Additionally, please explain how the enrollment process was applied. Clarifying these points will provide readers with a better understanding of your study's scope and sample representativeness.

The volunteers were recruited through posters in our laboratories and at the Paris-Saclay University. This has been added in the Method.

I recommend moving the inclusion and exclusion criteria into a separate subsection under the methods section. This will improve the organization of the methods and make it easier for readers to follow your experimental design.

This has been done.

The reporting of results is somewhat unclear in its current form. For each result, please specify the type of data analysis conducted and ensure that all relevant statistics are provided. This should include effect sizes, which are currently reported only for some analyses. Including consistent effect size reporting across all analyses will help to provide a clearer interpretation of the results.

We now added the effect sizes across all analyses.

Finally, please include a Limitations subsection within the discussion. This will allow you to acknowledge any potential weaknesses in the study, such as sample size limitations or methodological constraints, and demonstrate transparency in the interpretation of the findings.

We have now included a Limitations paragraph in the discussion.